# Therapeutic Effect of Erythropoietin on Alzheimer’s Disease by Activating the Serotonin Pathway

**DOI:** 10.3390/ijms23158144

**Published:** 2022-07-24

**Authors:** Kyu-Ho Shim, Sungchan Ha, Jin Seung Choung, Jee In Choi, Daniel Youngsuk Kim, Jong Moon Kim, MinYoung Kim

**Affiliations:** 1Department of Biomedical Science, CHA University School of Medicine, Seongnam 13496, Korea; kyuhoshim@gmail.com (K.-H.S.); tjdcks6101@naver.com (S.H.); choungjs@gmail.com (J.S.C.); 2Rehabilitation and Regeneration Research Center, CHA University School of Medicine, Seongnam 13496, Korea; cji-012@daum.net (J.I.C.); jmkim1013@gmail.com (J.M.K.); 3Research Competency Milestones Program (RECOMP) of School of Medicine, CHA University, Seongnam 13496, Korea; dkim304@chauniv.ac.kr; 4Department of Rehabilitation Medicine, CHA Bundang Medical Center, CHA University School of Medicine, Seongnam 13496, Korea

**Keywords:** therapeutics, erythropoietin, serotonin, Alzheimer’s disease, amyloid beta-peptides, cognitive dysfunction

## Abstract

Alzheimer’s disease (AD) is a neurodegenerative disease characterized by memory impairment in patients. Erythropoietin (EPO) has been reported to stimulate neurogenesis. This study was conducted to determine the regenerative effects of EPO in an AD model and to assess its underlying mechanism. Recombinant human EPO was intraperitoneally administered to AD mice induced by intracerebroventricular Aβ oligomer injection. Behavioral assessments with novel object recognition test and passive avoidance task showed improvement in memory function of the EPO-treated AD mice compared to that of the saline-treated AD mice (*p* < 0.0001). An in vivo protein assay for the hippocampus and cortex tissue indicated that EPO treatment modulated neurotransmitters, including dopamine, serotonin, and adrenaline. EPO treatment also restored the activity of serotonin receptors, including 5-HT4R, 5-HT7R, and 5-HT1aR (*p* < 0.01), at mRNA levels. Furthermore, EPO seemed to exert an anti-inflammatory influence by downregulating TLR4 at mRNA and protein levels (*p* < 0.05). Finally, an immunohistochemical assay revealed increments of Nestin(+) and NeuN(+) neuronal cells in the CA3 region in the EPO-treated AD mice compared to those in the saline-treated AD mice. The conclusion is that EPO administration might be therapeutic for AD by activating the serotonergic pathway, anti-inflammatory action, and neurogenic characteristics.

## 1. Introduction

Alzheimer’s disease (AD) is the most common form of dementia in older adults [1]. It involves a slow and irreversible decline in cognitive function [2], associated with neuronal degeneration [3]. AD manifests itself as two core pathologies: accumulation of β-amyloid (Aβ) plaques and neurofibrillary tangles. Accumulated Aβ plaques and neurofibrillary tangles induce various types of neurodegeneration and augment inflammatory response in a brain with AD [4,5]. However, despite the efforts and investigations, there is no effective medicine for this disease.

Erythropoietin (EPO) therapy might have therapeutic potential in AD and needs to be evaluated not only for efficacy but also for its action mechanism. In a previous study, EPO administration attenuated the activation of microglia, a maker of neuroinflammatory status, and restored synaptic loss in an AD mouse model [6]. Preclinical models of AD indicate that EPO displays neuroprotective property in vivo by decreasing neuroinflammation [7] and increasing neurogenesis [8]. In both transgenic and Aβ-treated mouse models of AD, EPO therapy brought about recuperation of associative learning memory [9,10]. EPO has been validated to have a significant role in promoting neurogenesis during brain development [11]. Furthermore, a recent study revealed proliferation in the neurogenic area by EPO administration even in advanced adulthood brain [12]. Thus, EPO treatment might be used for neurodegenerative diseases, such as AD, due to its neuroprotective property with neurogenic effects.

As for the influence on neurotransmitters, EPO administration prevented serotonergic fiber loss and maintained the density in murine spinal cord injury [13]. Another study suggested that serotonergic activation of EPO might contribute to EPO-mediated cell proliferation and neurogenesis in the mouse hippocampus [14]. Serotonin (5-hydroxytryptamine, 5-HT) is involved in higher brain functions, including cognition and emotional behavior, as well as in many important neurological processes, such as sleep, appetite, body temperature, pain, and motor activity, which are mostly affected in AD [15]. Several studies have reported a significant loss of 5-HT 4 receptors (5-HT4R) in hippocampal and cortical neurons in AD patients [15]. The activation of 5-HT4R was shown to benefit cognition in rodents and primates [16]. A preclinical study suggested that 5-HT7R agonists might be useful in treating memory impairments associated with aging or AD [17]. Acute and chronic inhibition of 5-HT1a post-synaptic receptors resulted in a decrease in the number of newly generated cells in the dentate gyrus [18]. On the contrary, compounds with a high affinity for 5-HT6R, such as 5-HT6R antagonists, were reported to improve cognitive function in AD patients [19]. Therefore, 5-HT4R, 5-HT7R, and 5-HT1a activation may be beneficial, but 5-HT6R stimulation may exacerbate AD symptoms. However, so far, no research on AD examined EPO therapy-induced serotonergic system activation and neurogenic response simultaneously.

In this study, we used AD model mice induced by intracerebroventricular (ICV) Aβ-42 oligomer injection and administered recombinant human EPO (rhEPO) for therapeutic purpose. To investigate the mechanism aroused by the EPO treatment, changes in neurotransmitters including 5-HT, inflammatory status, and neuroregenerative responses in vivo were assessed.

## 2. Results

### 2.1. EPO Improves Cognitive Memory Function in Aβ-Treated Mice

Seven days before treatment initiation (D–7), 8-week-old B6 mice were injected ICV with Aβ-42 oligomer to induce AD phenotype. Then, 500 IU/kg/day of rhEPO was administered to the Aβ-injected mice via the intraperitoneal (IP) route for 5 consecutive days (Figure 1A). The mice were divided into three groups according to interventions and compared. The Aβ+EPO group received rhEPO therapy after AD modeling, the Aβ+saline group received saline after AD modeling as a disease control, and the B6 normal group was treated as a normal control. Before each weekly behavior test, the body weight was measured to assess the effects of the treatment on body weight. The body weight of the B6 normal group showed a mild increase during the experiment period. The Aβ+saline and Aβ+EPO groups showed a similar tendency, and the three groups did not show any significant difference in weight at each measurement (Figure 1B). To assess the therapeutic effects of EPO on the Aβ-induced cognitive dysfunction and memory impairment, behavior tests, including the novel object recognition test (NORT) and the passive avoidance task were conducted, in a series (Figure 1C).

During the training session before AD induction, analysis of object exploration displayed no preference between two similar objects (data not shown). After induction of AD with Aβ, until D0, the recognition index (RI) showed a lowered exploration of novel objects (below 50% RI) in both Aβ+saline and Aβ+EPO groups compared to B6 normal mice (Figure 2A). This memory impairment was significantly restored by rhEPO administration, which was maintained from D7 to D28 with a superior RI in comparison with that of the Aβ+saline mice (*p* < 0.01) (Figure 2B). The passive avoidance task also revealed spatial memory deficit from D12 to D33 in the Aβ+saline group, with short step-through latency, while B6 normal control and Aβ+EPO groups showed similarly long latencies (*p* < 0.0001) (Figure 2C). This result indicates that rhEPO administration preserves the memory of the shock event on D5. EPO treatment seemed to prevent memory deficit until the last examination, on D33, with the preservation of lengthened step-through latency (*p* < 0.001) (Figure 2D). Eventually, these neurobehavioral results suggest that EPO exerted a positive influence on learning and memory enhancement in the animal model, while saline did not show such efficacy.

### 2.2. EPO Regulates Several Neurotransmitter Levels in the Brain, including the Serotonin Pathway

To further investigate whether EPO may regulate intracerebral neurotransmitters, we quantified the dopamine, serotonin, acetylcholine, and adrenaline concentrations in the brain tissue using ELISA assays (each group *n* = 4). Previous research has reported remarkable reductions in the concentrations of neurotransmitters, including glutamate and acetylcholine, in the hippocampi and neocortices of AD patients [20,21]. Dopamine, serotonin, adrenaline, and acetylcholine concentrations were measured in three groups of mouse brain tissue, dissected 4 weeks after EPO treatment (Figure 3). In both hippocampus and cortex, there were no differences in the levels of all quantified neurotransmitters between Aβ+EPO and B6 normal groups, while those of Aβ+saline showed lowered levels of serotonin in the hippocampus and the cortex and dopamine in the cortex. The adrenaline level showed a reversed pattern. Aβ+EPO mice exhibited increments in dopamine and serotonin levels and decrements in adrenaline in both hippocampus and cortex (*p* < 0.05) (Figure 3A–C,E–G). The level of acetylcholine was elevated by rhEPO administration only in the hippocampus of the AD model (Figure 3D). These results indicate the restoring effect of EPO treatment on neurotransmitter abnormality in AD mice, especially in terms of the serotonin level. Accordingly, as the next step, we decided to investigate the effects of EPO on the serotonergic pathway in the AD model, which is also expected to affect neurogenesis [22].

### 2.3. EPO Recovers Serotonergic Receptor Activities and the Neuroinflammatory Status at mRNA Levels in the Hippocampus

To determine whether EPO affects the serotonergic neuron system and neuroinflammation at the mRNA level, we performed a quantitative real-time PCR (qRT-PCR). Among the serotonin receptor genes known to be associated with AD in mouse hippocampus, *5-HT4*, *5-HT6*, *5-HT7*, and *5-HT1a* receptors were selected for the experiment. The agonists of *5-HT4R* and *5-HT7R* and the antagonist of *5-HT6R* have been reportedly beneficial in AD [23]. Similarly, in this study, the mRNA levels of *5-HT4R* and *5-HT7R* in the Aβ+EPO group were upregulated but the Aβ+saline group (*p* < 0.01) showed downregulation of these receptors on AD induction (*p* < 0.05) (Figure 4A,C). Moreover, EPO-treated mice downregulated the *5-HT6R* expression level, which was upregulated in the Aβ+saline group (*p* < 0.01) (Figure 4B). Receptor *5-HT1aR*, also reported to be associated with neurogenesis, significantly decreased in Aβ+Saline mice and increased in Aβ+EPO injected mice (*p* < 0.0001) (Figure 4D).

The mRNA levels of *TNF-α*, *NLRP3*, *TLR4*, and *FAS*, known as inflammation- and apoptosis-related factors, were also investigated. The mRNA levels of *TNF-α*, *NLRP3*, and *TLR4* were reduced in Aβ+EPO mice (*p* < 0.05) (Figure 4E–G), and *FAS* did not show a significant difference, although the trait was similar to that of the other molecules (Figure 4H). Experiments for the same factors as above in the cortex did not show significant results as in the hippocampus (data not shown).

Taken together, these results suggest that EPO treatment may have exerted therapeutic effects on cognition by modulating receptor activities of serotonin and reducing inflammation in the hippocampus of the AD mouse model.

### 2.4. EPO Promotes Neurogenesis and Suppresses Neuroinflammation at the Protein Level

To determine the effect of EPO on the brain at the protein level, Western blots were performed with prepared bilateral hippocampal and cortical samples (Figure 5A,F). As indicators of neurogenesis and neuroprotection, protein expression levels of Nestin (Figure 5B,G), NeuN (Figure 5C,H), and BDNF (Figure 5D,I) were quantified using each antibody. Intensities of the antibody bands significantly increased in samples of the Aβ+EPO group compared to those of the Aβ+saline group (*p* < 0.05), which were downregulated by AD induction with lower values than normal control (*p* < 0.05). According to these findings, the IP injection of rhEPO exerted neurogenic and neuroprotective effects on the mouse brain damaged by Aβ oligomer. In the hippocampus, the protein expression of TLR4, which was upregulated on Aβ administration, decreased on EPO treatment (*p* < 0.05) (Figure 5E). All protein levels were normalized to β-actin. The result also indicates that EPO ameliorates neuroinflammation in AD.

### 2.5. EPO Stimulates Neurogenesis in the CA1 and CA3 Areas of the Hippocampal Pyramidal Neuron Layer

To further evaluate the neuro-regenerative effect of EPO, the pyramidal neuron layer of the hippocampus was visually confirmed through cresyl violet staining. All coronal sections were 20 μm thick, and the cresyl violet staining time was the same. They were also photographed at the same light exposure, preference, and magnification. Micrographs were taken in areas CA1 and CA3 at 100× and 400× magnifications (Figure 6A,B). Three tissue slices were analyzed for each mouse, and four mice per group were prepared. The Nissl-stained area in the CA1 region of the Aβ+saline mice appeared slightly hazy and thin compared to the B6 normal (Figure 6A). On the contrary, CA1 of Aβ+EPO was observed to be thicker than that of Aβ+saline. This pattern was much more evident in the CA3 region (Figure 6B). The differences between each group were significant when the total areas of the CA1 and CA3 regions were measured using the ImageJ program after thresholding the same intensity (Figure 6C; CA1, *p* < 0.01; CA3, *p* < 0.005). These staining image results strongly suggest that EPO repairs Aβ-induced neuronal damage and induces neurogenesis.

## 3. Discussion

Erythropoietin (EPO), originally considered as an erythropoietic cytokine, is now known to have cytoprotective effects that have been established in several organs, particularly the brain [24]. EPO has been already used as a therapeutic product worldwide, which represents safety for clinical application. Experimental administration of exogenous EPO revealed amelioration of neuronal impairments related to several disease conditions, such as cerebral ischemia [25], intracerebral hemorrhage [26], and traumatic brain injury [27]. The present research proposed to confirm the beneficial effects of EPO in an AD mouse model and to investigate the underlying mechanisms. We could provide strong evidence that 5 consecutive days of rhEPO administration leads to neuroprotection against memory deficits induced by Aβ 42 oligomers and inflammation-related toxicity in the brain. On the basis of a previously conducted clinical trial, each single dose of rhEPO administered was determined to be 500 IU/kg [28]. Consequently, rhEPO protected against Aβ-induced neuronal damage at an IP dose of 500 IU/kg/day, corresponding to 2500 IU/kg in total.

As for its possible use in treating AD, EPO has been reported to produce a neurotrophic effect that reduces pathologic abnormalities, including Aβ accumulation, oxidative stress, neurodegeneration, and neuroinflammation and repairs nerve damage in AD [29]. In the present research, to evaluate possible efficacy of EPO treatment in AD, firstly, neurobehavioral outcomes were evaluated in Aβ-induced AD model mice. Two representative behavioral analyses were used: NORT, a long-term memory test, and passive avoidance task, a long-term event-space memory assessment [7]. The ability to recognize a novel object was significantly improved in AD mice after rhEPO injection, in contrast to saline-injected disease controls (Figure 2). The recognition ability of the EPO-injected group was also enhanced to a level similar to that of B6 normal mice and was maintained for 28 days after initiation of rhEPO injection. In addition, in a passive avoidance task that set the shock acquisition phase after rhEPO administration, the Aβ+EPO group responded to shock with retained memory similar to the B6 normal control group for up to four following weeks, while the Aβ+saline group showed short latencies in the light room consistently due to loss of memory (Figure 2C,D). These contrasting behavioral outcomes in Aβ+EPO and Aβ+saline indicate that EPO shows therapeutic effects by ameliorating memory dysfunction. This is consistent with previous reports showing prevention and reversion of Aβ-induced learning and memory deterioration by EPO treatment, which was assessed using behavioral Morris Water Maze (MWM) [30]. Consequently, these results support the idea of EPO as a potential therapeutic strategy for pathological memory deficits produced by AD.

To investigate the therapeutic mechanism of EPO on AD, we firstly assessed changes in cerebral neurotransmitters, including dopamine, serotonin, acetylcholine, and adrenaline [31]. Previous studies have shown that EPO has a trophic effect on dopaminergic neurons in a PD rat model [32], and also stimulates striatal dopamine release in the rat brain [33]. To our knowledge, no study directly revealed the effect of EPO treatment on concentration and receptors of serotonin in the brain, and this research could reveal serotonergic system activation by EPO in an AD model for the first time. Interestingly, the ELISA results showed that EPO administration enhances serotonin as well as dopamine (Figure 3A,B). According to a past postmortem AD study, 5-HT receptors were significantly decreased in many cases [34]. Monotherapy in AD using 5-HT agonists or various serotonin-mimetic compounds not only restored lowered levels of serotonin but also improved cognitive and behavioral disturbances [35]. On the basis of the results of this study, serotonin was speculated as one of main pathways that influence cognitive outcome in the EPO-administered Aβ-induced AD model.

To determine how rhEPO administration activates serotonin signaling, the research team confirmed the expression of each 5-HT receptor through qRT-PCR. Several types of 5-HT receptors were evaluated: the mRNA expressions of *5-HT4R*, *5-HT7R*, and *5-HT1aR* were found to be significantly elevated, and that of *5-HT6R* was depressed in the hippocampus of EPO-injected mice (Figure 4A–D). The results of changes in different subsets of 5-HT receptor and behavioral improvements seem to be in accord with previous reports that have revealed receptor-dependent effects for AD treatment. There was a report that 5-HT4R decreases in the hippocampus and cortex of AD patients [36]. Moreover, 5-HT4R agonists therapy was reported to increase BDNF, a neurogenesis marker, and to enhance synaptic plasticity, making it a potential therapeutic strategy for ameliorating memory, mood, and feeding behavior disturbances [16,37]. The administration of 5-HT7R agonists also reportedly contributed to improved memory formation [17]. As for 5-HT1aR, its antagonist was reportedly involved in inhibiting cell proliferation in adult mice dentate gyrus [18] and stimulating 5-HT1aR-affected adult hippocampal neurogenesis [38]. Therefore, 5-HT1aR could be involved in neuronal regeneration in the AD brain. In addition, 5-HT6R is an emerging marker for cognitive improvement in diseases of the central nervous system, which may vary in several neurotransmitter systems, depending on its antagonists, including serotonin [19]. Considering the tendency of serotonin receptors shown in this study, EPO seems to be involved in memory improvement and nerve regeneration through various serotonin receptors.

EPO might induce neurogenesis in neuropsychiatric disorders [12] and neurological diseases, such as stroke [39]. To examine the neurotrophic effects that could be associated with changes in various 5-HT receptor activities on EPO treatment of AD, we performed Western blotting (Figure 5). Various neuronal biomarkers, such as Nestin, NeuN, and BDNF, all increased in rhEPO-injected mice (Figure 5B–D,G–I). The results indicate possible EPO-induced neuroprotection and neurogenesis as these markers were used for confirming therapeutic effects in previous studies [22,40]. On the basis of a previous report indicating that the administration of the 5-HT4R agonist increases hippocampal cell proliferation and upregulates neuroplasticity-related markers [41], the increase in BDNF in these results can be considered as the effect of serotonin increase on EPO administration. These data suggest a possible novel relationship between EPO and serotonin. To visualize this neurogenic effect of EPO treatment, neuronal cell viability was investigated through neuronal staining. It was observed that the total Nissl-stained areas in CA1 (Figure 6A) and CA3 (Figure 6B) regions were diminished in Aβ+saline and preserved in Aβ+EPO (Figure 6C). It was reported that the density, volume, and thickness of hippocampal CA1 neurons are reduced in NeuN immunostaining of AD model mice [42]. In this study, it was confirmed that the neurons in the CA1 and CA3 regions, which were reduced in the Aβ-induced AD model, increased significantly in the EPO-treated mice to a level similar to that of the normal control group (Figure 6). This result indicates the restoring effects of EPO for Aβ-induced nerve damage in the brain through its regenerative effect, which accompanies the amelioration of AD symptoms.

Additionally, anti-inflammatory responses were observed as another candidate for 5-HTR mRNA activation mediating improvement in AD behavior. Increased immune responses were reported to cause nerve damage in an animal model of AD [42]. Chronic administration of 5-HT4 receptor agonists has been reported to reduce brain inflammatory processes associated with AD progression, such as amyloid pathology [23]. Several specific Toll-like receptors have been studied as various therapeutic targets in AD [43]. In particular, TLR4 was thought to mediate the neurotoxic action of DAMPs associated with AD-related neuronal injury [44]. Furthermore, the NLRP3 inflammasome, which cross-talks with TLR4, is also being actively studied as a mechanism of AD [45]. In this study, it was confirmed that EPO treatment downregulates mRNA expression of the inflammatory factors *NLRP3*, *TLR4*, and *TNF-α* (Figure 4F–H) and the apoptosis factor *FAS* (Figure 4E) induced by Aβ toxicity. EPO also repressed the protein expression level of TLR4 (Figure 5). These results suggest that EPO may induce anti-inflammatory effects, promoting recovery from the pathogenic mechanism of AD. In our experiments, these anti-inflammatory effects are considered to be related to cognitive improvement by EPO treatment.

This study has some limitations. Firstly, our experiments did not include evidence directly linking the serotonin pathway and the therapeutic effect of EPO. It only clearly confirmed that administered rhEPO improves cognitive ability and concurrent neurotrophic and anti-inflammatory responses. Further experiments should be conducted to clarify the connection between changes in the 5-HT receptor activities and neuronal recovering responses. Secondly, we used AD model mice in which Aβ 42 oligomers were injected ICV into wild-type C57BL/6 mice. Because the Aβ peptide was injected from the outside, Aβ plaques could not be assessed in this model. In addition, the generated Aβ-induced model was inevitably different from the transgenic mouse AD model, which might mimic AD pathology more. For clinical application as a neurotherapeutic agent in the future, it is necessary to verify the effect of rhEPO injection in a more reliable AD model. 

In conclusion, this study demonstrates the efficacy of rhEPO treatment in an AD model. Activation of the serotonergic system could be suggested as a main therapeutic mechanism of EPO therapy for the first time with neurotrophic and anti-inflammatory effects. It is also expected that identification of these mechanisms might be necessary in development of future therapeutics for AD. Further research revealing the precise mechanism of EPO on anti-inflammation and neurogenesis via serotonergic system in AD would provide a basis for therapeutic solution.

## 4. Materials and Methods

### 4.1. Animals

Male C57BL/6 mice, 6 weeks old, were purchased from KOATECH (Pyeongtaek, Gyeonggi-do, South Korea) and habituated for 14 days. Mouse housing and experiments took place within the CHA University animal facility. The mice were housed (5 animals per cage) in a temperature- and humidity-controlled facility on a 12 h/12 h light/dark cycle (lights on at 7:00 h), with ad libitum access to food and water. Behavioral experiments were executed between 9:00 h and 17:00 h. All animal procedures were conducted following a protocol approved by the Institutional Animal Care and Use Committee of CHA University (IACUC210116).

Aβ-induced AD mice were produced by stereotaxic injection of Aβ oligomer into the right lateral ventricle of 8-week-old male C57BL/6 mice. To acquire soluble oligomeric species, Aβ-42 (10 μM) [46] was incubated at 37 °C for 1 week after being dissolved in phosphate-buffered saline (PBS) with 10% dimethyl sulfoxide. Concisely, mice were anesthetized using the Vevo Compact Anesthesia System. Isoflurane was inducted by the same individual each time, with the nose of the mouse restrained into the stereotaxic apparatus. Then, 4 μL of Aβ oligomer was swiftly (1.0 μL/min) injected into the ICV space in the lateral ventricle. In this study, the bregma coordinates used for injection were −1.8 mm lateral, −1.0 mm posterior, and −2.5 mm below [46].

The Aβ-induced AD mice (*n =* 10) were treated by intraperitoneal (IP) injection with rhEPO, 500 IU/kg/day, and 1000 IU/250 μL (LG Chem Ltd., Gyeonggi-do, Republic of Korea) diluted in 0.9% saline or an equal volume of 0.9% saline alone as a sham-treatment disease control (*n =* 10), daily, for 5 consecutive days starting 7 days after Aβ injection (D0). Since untreated normal B6 mice were also observed as a negative control, there were three groups: B6 normal, Aβ+saline, and Aβ+EPO. The behavioral experiments were conducted as blind tests by two experimenters so that the experimental results were not biased.

### 4.2. Recognition Memory Assessment Using NORT

From the Aβ injection day (D7), NORT was carried out every 7 days up to D28 to assess hippocampus-dependent episodic memory [47]. Mice were relocated to the experimental room and left undisturbed in their home-cage for 10 min acclimatization in the new environment. During the training session, each mouse was placed for 10 min in the assay arena (an empty rectangle box made of plastic material), which consisted of two identical familiar objects (FOs) in it. The mouse was then sent back to the home-cage. The next day, each mouse was re-placed in the same assay arena as used for the training session for 3 min habituation. Thereafter, one of the FOs remained unvaried, while the other one was switched with a novel object (NO) for a 5 min recording session. In this session, object exploration was measured as below and interest for the NO was inferred by calculating the recognition index (RI), i.e., the ratio of NO/(FO+NO) [31]. The interest of the mice in the NO was evaluated as exploration, which was described as the time the mice spent sniffing the objects or touching them with their noses or forepaws.

### 4.3. Event-Space Memory Assessment Using the Passive Avoidance Task 

A passive avoidance applicating device was purchased from Panlab (LE872/LE10026/ShutAvoid, Barcelona, Spain) [48]. A passive avoidance task was initiated by giving shock 5 days after rhEPO injection (denoted as Shock, D5). The passive avoidance chamber consisted of two compartments, with a door between the two, light and dark, and the bottom of the dark area contained a shock generator. Before being administered a shock, each mouse was first placed in a light chamber. Thirty seconds later, the door between the light/dark compartments was opened so that the mouse could move into the dark chamber. The initial step-through latency was measured from the door opening, which was all similar among the groups. All groups entered the dark chamber within a maximum of 30 s. They can be identified by the dots at the shock acquisition point indicated on day 5 in Figure 2C. Once the mice entered the dark chamber, the door closed immediately and an electric shock was delivered under the feet (0.9 mA, 3 s, once) through the shock generator. After this first acquisition phase, the mice were taken to their home-cages and returned to the test chamber after 7 days. For the step-through latency phase (7-day delay) experiment, each mouse was placed into the light chamber again and step-through latency to cross through the door into the dark compartment was measured (max 300 s). When a mouse entered the dark chamber, the guillotine door was automatically closed and the program recorded the time in seconds. If the mouse did not enter the dark chamber after 300 s, the program was stopped and 300 s were recorded. Then, the step-through latency trials were delivered three more times, at 14-, 21-, and 28-day delay periods, to monitor the duration of the acquired memory.

### 4.4. Brain Tissue Preparation and ELISA Assay

Mice were euthanized in a CO_2_ chamber at 33 days after rhEPO administration. Their brains were immediately removed, and isolated brains were frozen in liquid nitrogen and stored at −80 °C until the hippocampus and the cortex were dissected. Each hippocampus and cortex from bilateral hemispheres were dissected on ice, frozen and stored at −80 °C until analysis. After they were thawed, the dissected hippocampus and cortex brain tissue were homogenized in 1× PBS buffer (100 μL and 200 μL, respectively). This homogenate was centrifuged at 10,000 rpm for 10 min, at 4 °C, and the supernatants were used for ELISA assays according to the manufacturer’s instructions. The commercial ELISA kits (LSBio, Seattle, WA, USA) used are summarized in Table 1. 

### 4.5. Western Blot Analysis

After using the supernatant in the above ELISA method, 80 μL of 1× PBS was added to the remaining pellets and the mixture vortexed several times to uniformly mix the homogenate. Some of these (hippocampus 70 μL and cortex 100 μL, respectively) were placed into 130 μL and 200 μL of RIPA Lysis and Extraction Buffer (Thermo Scientific™ #89900, Waltham, MA, USA) containing Phosphatase Inhibitor Cocktail 2 (Sigma-Aldrich #P5726, St. Louis, MO, USA), Phosphatase Inhibitor Cocktail 3 (Sigma-Aldrich #P0044, St. Louis, MO, USA), and Protease Inhibitor Cocktail 5 (quartett GmBH #QTPPI1015, Berlin, Germany). This lysate was vortexed every 30 min and lysed on ice for 2 h. The supernatants were obtained after centrifugation at 13,000 rpm for 20 min at 4°C. The supernatant samples were collected, and protein levels were quantified through Pierce™ BCA Protein Assay (Thermo Scientific™ #23225, Waltham, MA, USA) to prepare samples for Western blotting. The measured proteins were prepared so the final concentration was 20 μg/20 μL, using Tris-Glycine SDS Sample Buffer (2×) containing 10% β-mercaptoethanol. The samples were boiled in a water bath for 10 min. Protein samples were separated via 10% or 12% SDS-PAGE and then transferred to polyvinylidene fluoride membranes. Non-specific sites were blocked with 5% bovine serum albumin at room temperature for 1 h. The membranes were incubated with primary antibodies, at 4 °C, overnight. The blots were incubated with anti-rabbit or anti-mouse IgG conjugated to HRP (1:10,000 or 1:20,000, respectively) at room temperature for 1 h. Immunoreactive bands were visualized using an ImageQuant™ LAS 4000 (GE Healthcare, Chicago, IL, USA). The used antibodies are listed in Table 2.

### 4.6. Quantitative Real-Time PCR

After transferring a portion of the mixture using the Western blot method above, 700 μL of RNAiso Plus (TaKaRa Bio Inc. #9109, Shiga, Japan) was added to the residue in a 1.5 mL microtube. After sufficient vortexing, the microtube was placed on ice for 10 min. Then, 200 μL of chloroform was added and the microtubes were vortexed. They were centrifuged at 13,000 rpm for 20 min, at 4 °C, to extract only the upper layer, and we transferred them to a new tube containing 400 μL of isopropanol each. They were mixed by inverting gently several times and incubated, at −20 °C, for 30 min. After centrifugation at 13,000 rpm for 10 min, at 4 °C, as much of the supernatant as possible was removed and 600 μL of 70% ethanol (diluted in diethyl pyrocarbonate-treated water) was added for washing. After the mixture was centrifuged at 13,000 rpm for 10 min, at 4 °C, ethanol was removed and pellets were dried for 5 min. RNA was dissolved in 20 μL of DEPC water. RNA concentration was measured using SPECTROstar^®^ Nano (BMG LABTECH, Ortenberg, Germany). Then, DNA (2000ng/μL) was synthesized using Maxime™ RT PreMix Kit (iNtRON #25081, Seongnam, Gyeonggi-do, Korea) according to the manufacturer’s instructions. The synthesized DNA was used with AccuPower^®^ 2X GreenStar™ qPCR MasterMix (BIONEER #K-6254, Daejeon, South Korea) at a concentration of 500 ng/μL and measured with the CFX Connect Real-Time PCR Detection System (BIO-RAD #1855201, Hercules, CA, USA). Table 3 summarizes the primers used.

### 4.7. Tissue Preparation for Cresyl Violet Staining and Photomicrograph

The mice were euthanized in a CO_2_ chamber at 33 days after rhEPO administration. The mice were perfused with 30 mL of 1× PBS. The extracted brains were fixed in 4% paraformaldehyde for 24 h and then dehydrated in 30% sucrose solution. Brain samples were frozen with FSC 22 Clear Frozen Section Compound (Leica #3801480, Wetzlar, Germany) and 20 μm-thick coronal slices cut using a Leica CM1520 cryostat. Mounted frozen sections of positively charged slides were air-dried overnight and then placed in distilled water for washing. After being stained with 0.1% Cresyl violet solution prepared with Cresyl violet acetate (Sigma-Aldrich #C5042, St. Louis, MO, USA) for 4 min, the slides were rinsed with 1× PBS for 1 min. Then, they were dehydrated in 70%, 80%, 90%, and 95% ethanol solutions for 2 min each. They were cleared in xylene solution for 5 min and immediately mounted with the Permanent™ mounting medium (Fisher Chemical™ #SP15, Hampton, VA, USA). Photographs of the prepared slides were taken with the i-solution™ Auto Plus (IMT, Vancouver, BC, Canada) program using an ECLIPSE Ts2 (Nikon, Tokyo, Japan) instrument. The photos were taken at magnifications of 100× and 400×. Cresyl violet acetate solution stained the Nissl material in the cytoplasm of neurons, and the neuropil was stained with granular purple-blue. This stain was used to identify neural structures.

### 4.8. Statistical Analysis

To determine longitudinal differences in the behavior scores between the groups during the experimental period, all statistical analyses in this study were performed using two-way repeated measures analysis of variance (RM ANOVA, *F* value) in SPSS Statistics version 20 (IBM, Armonk, NY, USA). In addition, a two-tailed unpaired *t*-test in GraphPad Prism 8 (GraphPad Software, San Diego, CA, USA) was used to compare the behavior scores between the two groups at a time point. Statistical differences were considered to be significant when *p* < 0.05 (* *p* < 0.05, ** *p* < 0.01, *** *p* < 0.005, and **** *p* < 0.0001). Two-way analysis of variance followed by Tukey’s post hoc test was used for group comparisons. Data of ELISA, qRT-PCR, and Western blotting were obtained from a smaller number of samples and were not normally distributed. Accordingly, non-parametric *t*-tests were used for these data.

## Figures and Tables

**Figure 1 ijms-23-08144-f001:**
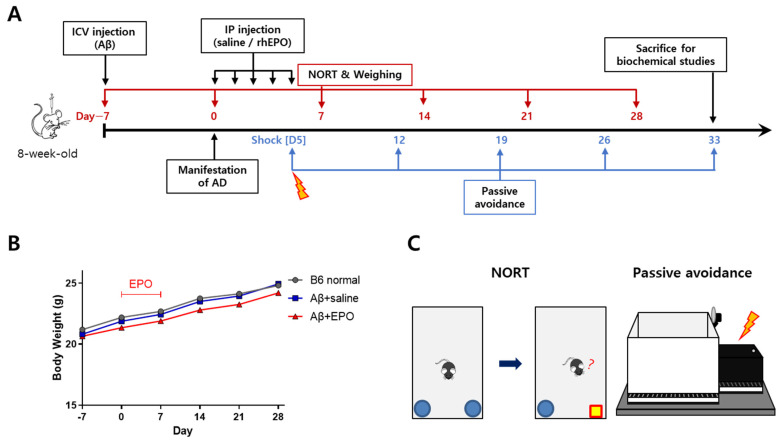
Erythropoietin (EPO) treatment and animal behavioral assessment design. (**A**) Whole in vivo experiment schedule: Beta amyloid-42 oligomer (Aβ) was injected into the intracerebroventricular (ICV) space at D7 (7 days before EPO treatment). Then, recombinant human EPO (rhEPO) or saline was injected daily via the intraperitoneal (IP) route for 5 consecutive days from D0. The novel recognition object test (NORT) was performed on a weekly basis, from D7 to D28. Body weight was measured right before every NORT. The passive avoidance acquisition phase (shock phase) was carried out 5 days after rhEPO injection. Then, step-through latency trials were performed 7, 14, 21, and 28 days after the shock phase to monitor the duration of the acquired memory from the shock. (**B**) Body weight gains of the three groups during the test periods were not significantly different (each group *n* = 8 to 10). (**C**) Depictions presenting two neurobehavioral assessment methods used in this study were: NORT and passive avoidance test.

**Figure 2 ijms-23-08144-f002:**
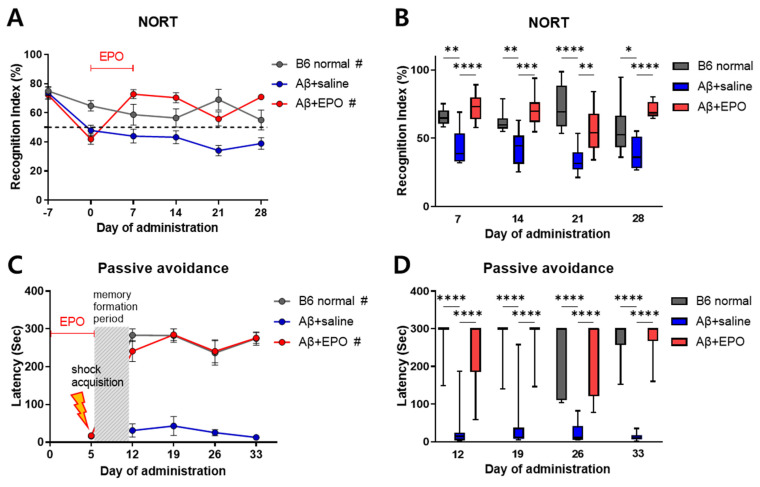
EPO protects against memory deterioration in Aβ-induced Alzheimer’s disease (AD) model mice. (**A**,**B**) long-term recognition memory function assessed with NORT and (**C**,**D**) long-term event-space memory assessment using a passive avoidance task in the AD mouse. Mice were analyzed for NORT on days −7, 0, 7, 14, 21, and 28. On day 5, they had acquisition phase for passive avoidance, and after retention duration, the mice were examined on days 12, 19, 26, and 33. *n* = 8 to 10 mice per group. Significant inter-group differences were found with # *p* < 0.0001 for B6 normal and Aβ+EPO groups compared to the Aβ+saline group according to two-way repeated-measures ANOVA in (**A**,**C**); * *p* < 0.05, ** *p* < 0.01, *** *p* < 0.005, and **** *p* < 0.0001 for B6 normal and Aβ+EPO groups compared to the Aβ+saline group according to independent *t*-test in (**B**,**D**).

**Figure 3 ijms-23-08144-f003:**
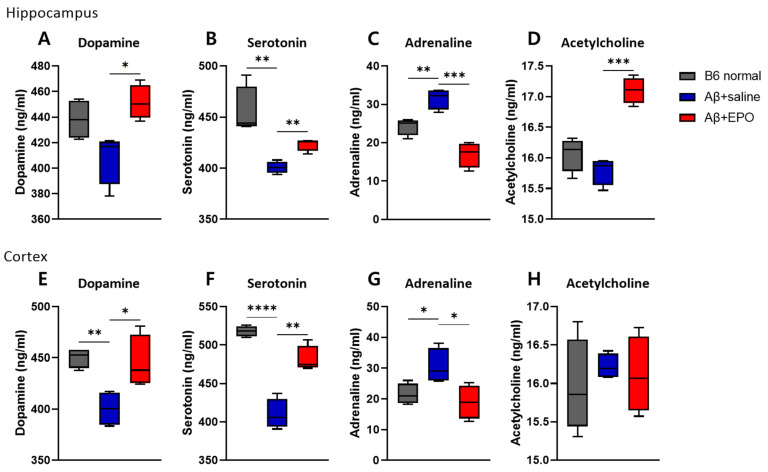
EPO regulates neurotransmitter levels in the brain. The administration of rhEPO influences the concentrations of several neurotransmitters, such as (**A**,**E**) dopamine, (**B**,**F**) serotonin, (**C**,**G**) adrenaline, and (**D**,**H**) acetylcholine, in both hippocampus and cortex tissues. *n* = 4 per group. * *p* < 0.05, ** *p* < 0.01, *** *p* < 0.005, and **** *p* < 0.0001 according to independent *t*-test in (**A**–**H**).

**Figure 4 ijms-23-08144-f004:**
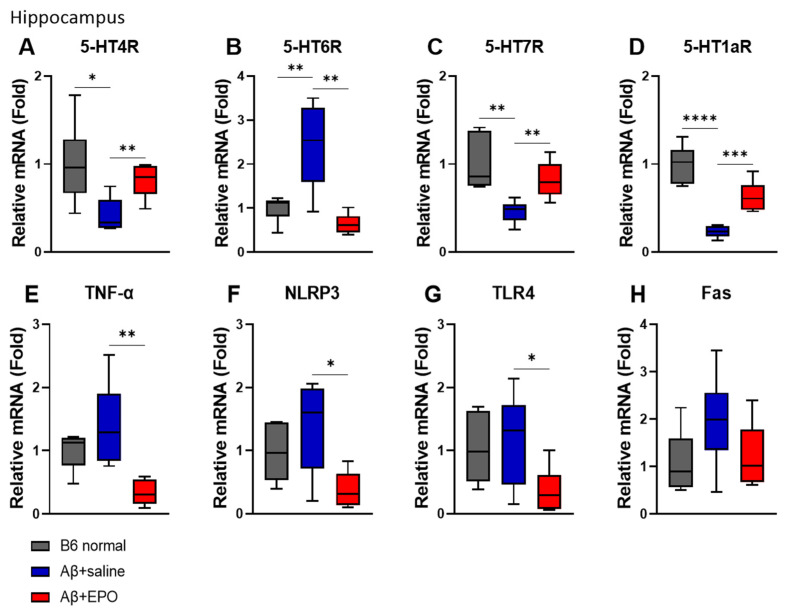
EPO regulates serotonin receptors and neuroinflammation at mRNA levels in the hippocampus. (**A**–**D**) Serotonin; *5-hydroxytryptamine (5-HT)* receptors and neuroinflammation-related factors, such as (**E**) *TNF-α*, (**F**) *NLRP3*, (**G**) *TLR4*, and (**H**) *FAS*, were regulated by EPO. *n* = 6 per group. * *p* < 0.05, ** *p* < 0.01, *** *p* < 0.005, and **** *p* < 0.0001 according to independent *t*-test in (**A**–**H**).

**Figure 5 ijms-23-08144-f005:**
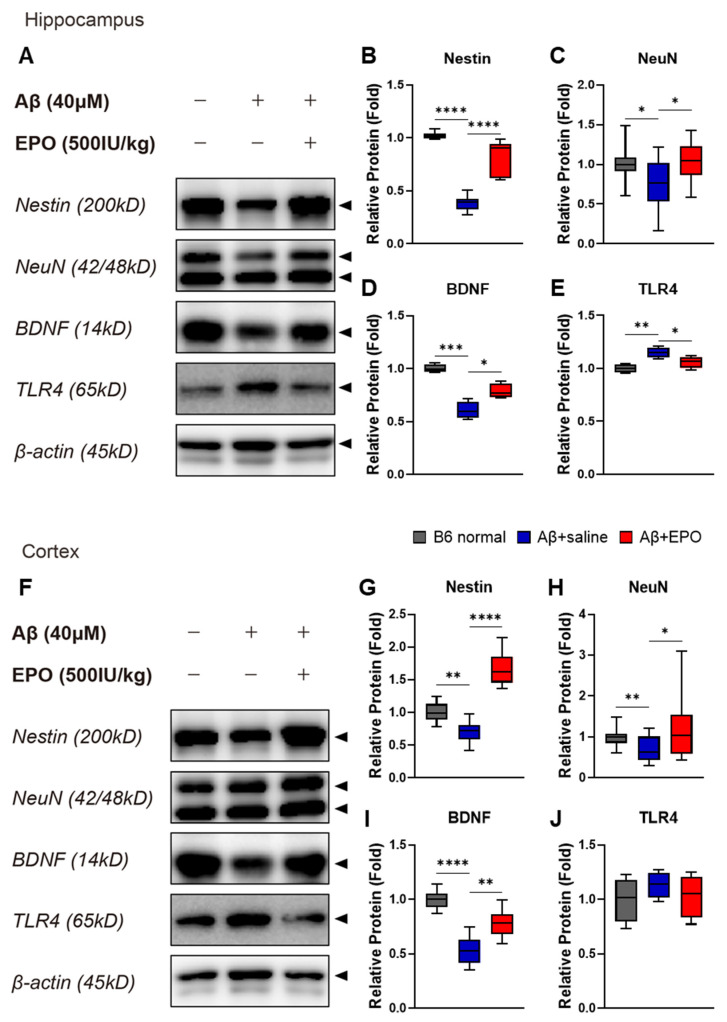
EPO promotes neurogenesis and suppresses neuroinflammation at the protein expression level. Western blotting band images (**A**,**F**; arrow heads) and relative protein expression level bars (**B**–**E**, **G**–**J**) display the effect of rhEPO in AD mice. Treatment with rhEPO significantly increased expression levels of (**B**,**G**) Nestin, a neuronal precursor cell marker, (**C**,**H**) NeuN, a neuronal cell biomarker, and (**D**,**I**) BDNF, a neurotrophic factor, both in the hippocampus and the cortex. (**E**,**J**) The expression of TLR4, known as a neuroinflammatory factor, was decreased in the hippocampus in the Aβ+EPO group compared to that in the Aβ+saline group. *n* = 4 per group. * *p* < 0.05, ** *p* < 0.01, *** *p* < 0.005, and **** *p* < 0.0001 according to independent *t*-test in (**B**–**E**, **G**–**J**).

**Figure 6 ijms-23-08144-f006:**
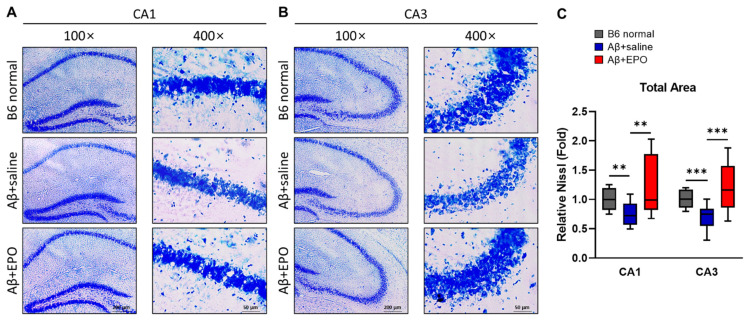
Neuro-regenerative effect of EPO on Aβ-induced dementia model mice visualized with pyramidal cell density in the CA1 and CA3 areas of the pyramidal neuron layer within the mouse hippocampus. (**A**,**B**) Representative photomicrographs of coronal sections of hippocampal CA1 and CA3 subfields stained with 0.1% cresyl violet solution were taken at 100× and 400× magnifications. All the samples were stained simultaneously to maintain an identical condition. The images were taken with the same amount of light and at the same exposure time and magnification. (**C**) Chart shows total area of relative Nissl-stained levels. Three slices were measured per mouse, and there were 4 mice per group. ** *p* < 0.01 and *** *p* < 0.005 according to independent *t*-test in (**C**).

**Table 1 ijms-23-08144-t001:** ELISA products used in this study.

Target	ELISA Product	Manufacturer, #Catalog No.
Dopamine	Mouse Dopamine (Competitive EIA) ELISA Kit	LSBio, Seattle, WA, USA #LS-F28308
Serotonin	Mouse Serotonin (Competitive EIA) ELISA Kit	LSBio, Seattle, WA, USA #LS-F28406
Adrenaline/Epinephrine	Mouse Adrenaline / Epinephrine (Competitive EIA) ELISA Kit	LSBio, Seattle, WA, USA #LS-F28134
Acetylcholine	Mouse Acetylcholine (Competitive EIA) ELISA Kit	LSBio, Seattle, WA, USA #LS-F28199

Abbreviations: EIA, enzyme immunoassay; ELISA, enzyme-linked immunosorbent assay.

**Table 2 ijms-23-08144-t002:** Antibodies used in this study.

Target Protein	Name of Antibody	Host Animal	Manufacturer, #Catalog No.	Dilution for Western Blot
Nestin	Nestin antibody [2Q178]	Mouse, monoclonal	Abcam, Cambridge, England #ab6142	1:1000
NeuN	NeuN antibody	Rabbit, polyclonal	Merck Millipore, Darmstadt, Germany #ABN78	1:1000
BDNF	BDNF antibody	Rabbit, polyclonal	Invitrogen, Waltham, MA, USA #PA5-85730	1:1000
TLR4	TLR4 antibody	Rabbit, polyclonal	Novus Biologicals, Centennial, CO, USA #NBP1-78427	1:1000
β-actin	β-actin (C4) antibody	Mouse, monoclonal	Santa Cruz, Dallas, TX, USA #sc-47778	1:1000
Rabbit	rabbit IgG, HRP-linked antibody	Goat	Cell Signaling, Danvers, MA, USA #7074	1:10,000
Mouse	mouse IgG, HRP-linked antibody	Horse	Cell Signaling, Danvers, MA, USA #7076	1:20,000

Abbreviations: BDNF, brain-derived neurotrophic factor; TLR4, Toll-like receptor 4; HRP, horseradish peroxidase.

**Table 3 ijms-23-08144-t003:** Primers used in this study.

Gene	Gene Bank Number	Primer	Sequence (5′–3′)	Annealing Temperature (°C)	Product Size (bp)
5-HT4R	NM_008313	Forward	TGCTCACGTTCCTTGCAGTGGT	60	134
		Reverse	GTCAGCAAAGGCGAGAGACACA		
5-HT6R	NM_021358	Forward	TGCCATCTGCTTCACCTACTGC	60	139
		Reverse	CTACTGTCAGCAGACTCCATCC		
5-HT7R	NM_008315	Forward	TCATGACCCTGTGCGTGATCAG	60	119
		Reverse	GAGAAGCCAGACCGACAGAATC		
5-HT1aR	NM_008308	Forward	TGCCAACTATCTCATCGGCTCC	60	103
		Reverse	CAGAGTCCACTTGTTGAGCACC		
FAS	NM_007987	Forward	TATCAAGGAGGCCCATTTTGC	60	195
		Reverse	TGTTTCCACCTCTAAACCATGCT		
NLRP3	NM_004895	Forward	GGACTGAAGCACCTGTTGTGCA	60	153
		Reverse	TCCTGAGTCTCCCAAGGCATTC		
TLR4	NM_138554	Forward	CCCTGAGGCATTTAGGCAGCTA	60	126
		Reverse	AGGTAGAGAGGTGGCTTAGGCT		
TNF	NM_013693	Forward	CCAACGGCATGGATCTCAAAGACA	60	141
		Reverse	AGATAGCAAATCGGCTGACGGTGT		

Abbreviations: HTR, hydroxytryptamine (serotonin) receptor; NLRP3, NOD-like receptor pyrin domain-containing protein 3; TLR4, Toll-like receptor 4; TNF, tumor necrosis factor.

## Data Availability

The data presented in this study are available upon request from the corresponding author.

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
