# Peer review of "Therapeutic Effect of Erythropoietin on Alzheimer’s Disease by Activating the Serotonin Pathway"

_ijms, 2022, doi:10.3390/ijms23158144_

Round 1
Reviewer 1 Report
This paper has screened the Erythropoietin neurogenesis effect in a mouse administered with Abeta oligomers through an intracerebroventricular route. This study has confirmed the previous results reported in several studies and shown results of erythropoietin effect on the levels of the different neurotransmitters. Point to point analysis of the study is below.
1. In figure 2 A NORT study, the authors need to confirm if there is no significant difference between the B6 normal and Abeta+ saline group.
2. In Figures 2 B and D authors need to include B6 normal and compare it with the Abeta+saline group.
3. Throughout the manuscript authors need to include band molecular weight in western blot. Because few blots were having two bands and it may create ambiguity for the readers.
4. In the introduction and discussion authors need to mention the novelty of the study compared to the previously reported erythropoietin neurogenesis studies.
Reviewer 2 Report
The manuscript "Therapeutic Effect of Erythropoietin on the Serotonin Pathway 2 in an Animal Model of Alzheimer's Disease" is a search for new therapies for Alzheimer's disease. My comments: Why did the authors choose Erythropoietin, which is not an inert agent. How do the authors want to use this therapy in the future? At what age were the animals used in the experiments. What were the control groups? Was an untreated control group and saline administered?
